# Towards Automation in Radiotherapy Planning: A Deep Learning Approach for the Delineation of Parotid Glands in Head and Neck Cancer

**DOI:** 10.3390/bioengineering11030214

**Published:** 2024-02-24

**Authors:** Ioannis Kakkos, Theodoros P. Vagenas, Anna Zygogianni, George K. Matsopoulos

**Affiliations:** 1Biomedical Engineering Laboratory, National Technical University of Athens, 15773 Athens, Greece; tpvagenas@biomed.ntua.gr (T.P.V.); gmatsopoulos@biomed.ntua.gr (G.K.M.); 2Radiation Oncology Unit, 1st Department of Radiology, ARETAIEION University Hospital, 11528 Athens, Greece; annazygo1@yahoo.gr

**Keywords:** head and neck cancer, CT, parotid glands, artificial intelligence, deep learning, radiation therapy, segmentation, registration

## Abstract

The delineation of parotid glands in head and neck (HN) carcinoma is critical to assess radiotherapy (RT) planning. Segmentation processes ensure precise target position and treatment precision, facilitate monitoring of anatomical changes, enable plan adaptation, and enhance overall patient safety. In this context, artificial intelligence (AI) and deep learning (DL) have proven exceedingly effective in precisely outlining tumor tissues and, by extension, the organs at risk. This paper introduces a DL framework using the AttentionUNet neural network for automatic parotid gland segmentation in HN cancer. Extensive evaluation of the model is performed in two public and one private dataset, while segmentation accuracy is compared with other state-of-the-art DL segmentation schemas. To assess replanning necessity during treatment, an additional registration method is implemented on the segmentation output, aligning images of different modalities (Computed Tomography (CT) and Cone Beam CT (CBCT)). AttentionUNet outperforms similar DL methods (Dice Similarity Coefficient: 82.65% ± 1.03, Hausdorff Distance: 6.24 mm ± 2.47), confirming its effectiveness. Moreover, the subsequent registration procedure displays increased similarity, providing insights into the effects of RT procedures for treatment planning adaptations. The implementation of the proposed methods indicates the effectiveness of DL not only for automatic delineation of the anatomical structures, but also for the provision of information for adaptive RT support.

## 1. Introduction

Radiation therapy (RT) stands as the primary treatment for head and neck (HN) cancer, either utilized independently or in conjunction with adjuvant treatments such as surgery and/or chemotherapy [1]. Despite its effectiveness in cancer treatment, RT is not without challenges, as even the partial irradiation of healthy cells can inadvertently damage normal tissues (organs at risk, OARs), leading to complications [2]. Despite its efficacy in cancer treatment, RT presents challenges, given that even the partial irradiation of healthy cells can inadvertently harm normal tissues, referred to as Organs At Risk (OARs), thereby leading to complications [3]. Typically, physicians manually perform the determination of tumor and OAR boundaries, utilizing Computed Tomography (CT) images. However, this is a time-consuming and labor-intensive process (since radiologists have to work slice by slice), making any adjustments in treatment planning particularly challenging [4]. This is further exacerbated due to repeated and swift adaptations required on the treatment plan, particularly in cases that present 20–30% tumor-related volumetric changes between RT sessions [5,6]. On this premise, adaptive RT necessitates frequent updates of OARs and target volumes during treatment, making it imperative to delineate these structures quickly and efficiently.

The advent of artificial intelligence (AI) and machine learning has revolutionized the assessment of RT treatment, ranging from computer-aided detection and diagnosis systems to predicting treatment alteration requirements [7,8]. However, many studies still involve manual segmentation of parotid glands combined with machine learning for predicting the necessity of RT replanning, which could be further automated [9,10,11].

Specifically, in automatic segmentation, machine learning has proven very effective in contouring OARs within a reasonable timeframe, enhancing the safety and efficiency of treatments [12,13,14]. Despite this, producing atlases for the segmentation task can reduce the model’s ability to generalize on images from different patients due to the large anatomical difference, because an atlas may not be able to represent all variations occurring from the different anatomies and acquisitions [15]. Recent advancements in deep learning (DL) have shown superior performance in auto-segmentation, offering better feature extraction, capturing local relationships, reducing observer variation, and minimizing contouring time [16,17,18,19]. However, these studies suffer from limitations that can impede their usage and applicability. Although interactive segmentation can reduce time and errors compared to manual segmentation, fully automatic segmentation can achieve both at a greater level. In addition, 2D or slice-based segmentation cannot capture the 3D anatomical features, which are essential to enhance the DL model’s accuracy. The scarcity of labeled data is another challenge. DL models typically require a large amount of annotated data for training, and obtaining a comprehensive dataset with accurately annotated parotid glands in diverse clinical scenarios can be challenging. Moreover, the potential for class imbalance, where certain anatomical structures or pathologies are underrepresented in the dataset, may affect the model’s performance, leading to biased results [20].

Several recent studies have evaluated DL-based methods for auto-segmentation, showing promising results. For instance, Gibbons et al. [21] evaluated the accuracy of atlas-based and DL-based auto-segmentation methods for critical OARs in HN, thoracic, and pelvic cancer. Their results showed significant improvements in DL-based segmentation compared to atlas-based segmentation both on Dice Similarity Coefficient (DSC) and Hausdorff Distance (HD) metrics, while contour adjustment time was significantly improved compared to manual and atlas-based segmentation. In a similar manner, a recent study [22] compared the automatically generated OAR delineations acquired from a deep learning auto-segmentation software to manual contours in the HN region created by experts in the field. The software provided good delineation of OAR volumes, with most structures reporting DSC values higher than 0.85, while the HD values indicated a good agreement between the automatic and manual contours. Furthermore, Chen et al. [23] employed a WBNet automatic segmentation method and compared it with three additional algorithms, utilizing a large number of whole-body CT images. They reported an average DSC greater than or equal to 0.80 for 39 out of 50 OARs, outperforming the other algorithms. Furthermore, Zhan et al. [24] utilized a Weaving Attention U-net deep learning approach for multi-organ segmentation in HN CT images. Their method integrated convolutional neural networks (CNNs) and a self-attention mechanism and was compared against other state-of-the-art segmentation algorithms, showing superior or similar performance. As such, it was able to generate contours that closely resembled the ground truth for ten OARs, with DSC scores ranging from 0.73 to 0.95. Taking the above into consideration, it is evident that DL-based segmentation methods have the potential to enhance RT treatment for HN cancer patients by improving the precision of organ at risk delineation. However, it should be noted that DL algorithms are data-driven methods whose results can be influenced by sample imbalance, variance among different patients and anatomical regions, and observer bias. To address these, recent studies apply interconnected networks, while employing larger datasets from different hospitals to ensure its effectiveness across different patient populations [25,26,27].

Our study builds on these advancements, employing an AttentionUNet-based DL segmentation method for HN CT images. This method focuses on targeted structures, suppressing non-related structures to reduce false positives. We conducted a comprehensive evaluation on three datasets, including two public and one private dataset, comparing our method with state-of-the-art DL frameworks for automatic segmentation. The results indicate the high performance of our automatic segmentation approach, outperforming other DL methods significantly. As an extension of our segmentation framework, we also investigated the efficiency of our model in RT replanning based on volumetric changes in the parotid gland. Our registration framework, aligning the planning CT with the Cone Beam CT (CBCT) scans from different weeks, demonstrated the validity of the registration schema. The mean volumetric percentage difference in tumor volume between the first and fifth week of treatment sessions, when comparing automated registration with the ground truth, showed very low values, confirming the effectiveness of our framework in cases requiring replanning during RT or between RT sessions. Overall, the contributions of this study can be summarized as follows:A fast and automatic DL segmentation method based on the AttentionUNet is utilized for the segmentation of the parotid glands from HN CT images. The AttentionUNet-based method achieves both a reduction in false positive regions and the enhancement of the parotid glands’ segmentation.An extensive evaluation of the segmentation method was performed on two public datasets and one private dataset with varying acquisition parameters. Both qualitative and quantitative results support the model’s ability to generalize effectively in images with anatomical and imaging variability.A registration-based framework was designed and implemented to transfer the segmentation masks produced by the AttentionUNet model to the CBCT images of the following weeks. The results indicate the effectiveness of our framework in cases where replanning is required during or between RT sessions.

The subsequent sections of the paper are organized as follows. Section 2 provides a comprehensive overview of the datasets and the DL methodological framework for segmentation and registration. In Section 3, the results of the proposed schema are separately presented for segmentation and registration. The implications of our methods and results are discussed in Section 4. The entire paper is summarized in Section 5.

## 2. Materials and Methods

### 2.1. Datasets

#### 2.1.1. Public Dataset

Two public datasets were combined to evaluate the segmentation accuracy of the parotid glands. The first dataset is the PDDCA dataset [28], which consists of 48 head and neck CT images from the Radiation Therapy Oncology Group (RTOG). Images are accompanied by manual segmentation masks from several organs at risk (brainstem, mandible, chiasm, bilateral optic nerves, bilateral parotid glands, and bilateral submandibular glands). From these masks, we utilized only the two parotid glands, which are the target of this investigation. The reconstruction matrix for the data was 512 × 512 pixels with in-plane spacing from 0.76 × 0.76 to 1.27 × 1.27 mm. The spacing of the z dimension ranged from 1.25 mm to 3 mm and the number of slices ranged between 110 and 190.

The second dataset is HaN-Seg, the HN OAR CT and MR Segmentation Dataset [29], with 42 head and neck CT and MRI images from patients included in the publicly available training set (set 1). The dataset includes masks of 30 OARs delineated from the CT images, where we selected the parotid glands. CT scans were conducted using either the Philips Brilliance Big Bore from Philips Healthcare, Netherlands or the Siemens Somatom Definition AS scanner from Siemens Healthineers, Germany. Standard clinical protocols were employed, with X-ray tube voltage set between 100 and 140 kV, and the field of view (FoV) (500 × 500) − (800 × 800) measured in mm^2^. The matrix size ranged between 512 × 512 and 1024 × 1024, while the number of slices varied from 116 to 323. In-plane spacing ranged from 0.52 to 1.56 and z-dimension spacing from 2.0 to 3.0 mm.

Both datasets included images with different spacing values, a fact that introduced image variability and made the subsequent segmentation more challenging. On this premise, the two datasets were unified and a 5-fold cross-validation technique was applied. During the 5-fold cross-validation, we ensured that in each fold, samples from both datasets were included to test the model’s effectiveness on heterogenous images regarding size and number of slices.

#### 2.1.2. Private Dataset

The private dataset comprised planning CT (pCT) image data, collected from 23 patients (14 male) with squamous cell carcinoma of the HN from the ATTIKON University Hospital, Athens (Protocol EBD304/11-05-2022, 7 June 2022). Each pCT image included 87 to 197 slices with a size of 512 × 512 pixels for each slice and a 3 mm interslice distance. The private dataset also included images with a variable number of slices.

### 2.2. Data Preprocessing

Prior to the DL model application, a series of preprocessing steps were performed in order to reduce the effect of variability of CT images in the datasets and enhance the neural network’s training progress. In order to transform all images into a common spatial setting, images were resampled to a certain spacing, where the in-place spacing was set to 0.87 mm × 0.87 mm and the spacing in z dimension to 3 mm. The spacing values were selected by considering the mean and median spacing values. The previously mentioned resampling was also utilized, since both the in-plane spacing and the interslice spacing were variable across the images of the datasets. Linear and nearest neighbors’ interpolation methods were applied to images and label masks, respectively. Normalization and clipping were applied to reduce outlier values and assist the network to be trained. Hounsfield units (HU) in the CT images were rescaled and clipped to the range (−160, 240) HU, which is more suitable for the detection of the parotids corresponding to soft tissues and neighboring areas [30].

### 2.3. Segmentation Model

The main neural network architecture of the segmentation implemented for the delineation of the parotid glands from the pCT images was the AttentionUNet model [31]. The AttentionUNet-based architecture follows a UNet-like structure with the attention gates at the skip connections. Below, the attention gate and the main architecture are presented.

#### 2.3.1. Attention Gate

The attention gate enables the network to focus only on the most important structures for the target task. It can also reduce false positive predictions and improve accuracy [31]. To achieve this, feature maps from the encoder path pass through skip connections, which include attention gates (Figure 1).

In the attention gate block, both the gating signal (g) and input features (x) are linearly transformed by a convolutional layer with a kernel size of 1 (which is performed in a channel-wise manner). In this way, aligned weights are amplified and unaligned weights are reduced. The summation of the outputs of the two convolutions passes through a ReLU activation function. Subsequently, the signal is linearly transformed by a new convolution with a kernel size of 1. The following sigmoid layer is used to normalize the output, and consequently the weights, and assist convergence. Finally, a grid resampling technique using trilinear interpolation is applied to produce attention coefficients based on the spatial information of each point in the grid. In this procedure, attention aggregates information from multiple scales. The output of the attention gate is calculated by the product of the input features (x) and the attention coefficients.

#### 2.3.2. AttentionUNet Architecture

The proposed neural network architecture follows the UNet structure, consisting of an encoder, a bottleneck layer, and a decoder path (Figure 1). In the encoder path, each block includes two successive convolutional blocks and a downsampling layer in order to reduce the dimensions of the feature maps by 2 in each step. Each convolutional block comprises a convolutional layer with the corresponding number of filters, a batch normalization layer, and a ReLU activation function. Batch normalization standardizes the output of the convolutional layers, by ensuring that the mean and standard deviation are constant across mini-batches, and makes training more stable. The downsampling layer refers to a max pooling operation with a stride of 2, aiming to reduce the spatial information (and thus the memory requirements), while extracting the most important context information. After each downsampling step, the number of filters is doubled. The bottleneck layer also consists of two consecutive convolutional blocks. In our current design, the encoder layers include convolutions with 16, 32, 64, and 128 filters, respectively. The convolutional blocks of the bottleneck have 256 filters. The bottleneck layer constitutes the middle layer, which connects the encoder path with the decoder. The process of compression (which takes place in the encoder) was completed in the bottleneck layer, where the high-level features extracted from the input image capture the contextual information essential for segmentation. During the encoder’s layers, the context information to be extracted is increased, while the spatial information becomes coarser. This is achieved by the application of sequential convolutional blocks, enlarging the effective receptive field. On the contrary, the decoder path aims at reconstructing the output from the input feature maps. In this regard, at the end of each decoder layer, an upsampling technique (based on Transpose Convolutions) is applied. Transpose convolutions enable the upsampling of the feature maps from the previous layer of the decoder by inserting trainable parameters. In consensus with the common UNet structure, skip connections are also utilized to transfer information from the encoder to the decoder. However, attention gates are inserted into the skip connections to suppress irrelevant feature responses and extract the most important regions. By applying the attention gates before concatenation, the network focuses only on the most useful activations. In each decoder block, the concatenation of the output of the attention gate in the skip connection and the upsampled output of the previous layer are followed by a simple convolutional layer to merge the feature maps and half the number of feature maps. In the final layer, a convolutional layer with the number of filters equal to the number of classes and a softmax layer to extract the predictions for the segmentation are implemented. The details on the size of the feature maps across the proposed architecture are presented in Table 1 and Table 2.

In Table 1, the input and output sizes of the main blocks of the encoder path are presented in detail. The example input corresponds to the size of windows utilized during this study, which was [96, 96, 96]. As previously mentioned, in each block of the encoder, the dimensions of the feature maps are downsampled by 2 and concurrently the number of features is doubled. In the decoder path, the input and output sizes are omitted (since they are the same as the encoder but in reverse order).

In Table 2, the input and output sizes of the main blocks of the decoder are presented. In each one of the first 4 blocks of the decoder, the feature maps are upsampled by 2 and the number of filters is reduced by 2. The last layer utilizes 3 filters for the 3 classes (background, left parotid gland, right parotid gland).

#### 2.3.3. Evaluation Scheme and Loss Function

To assess the model’s performance, a 5-fold cross-validation methodology was employed on the two created datasets, the public and the private one. For the unified public dataset (consisting of the two public subsets), the folds were created by proportionally splitting the images from the two subsets in order to include images from both of them in the training and validation sets. For ground truth, the segmentation masks from parotid glands (annotated manually by experts) were utilized. For the segmentation task, we used masks with three labels, where the values 1 and 2 correspond to the left and right parotid and the value 0 to the background. In the evaluation, the metrics were calculated for the united parotid glands’ masks to provide a more elucidated perspective.

As such, the segmentation methodology evaluation included the Dice Similarity Coefficient (DSC), Hausdorff Distance (HD) metric, Recall, and Precision [32,33]. Briefly, the DSC coefficient quantifies the overlap between the ground truth mask and the predicted mask, while HD measures the spatial distance between the ground truth and the produced segmentation mask and provides an evaluation for the boundary delineation accuracy. For the segmentation evaluation, the mean value of DSC, Precision, and Recall was estimated, while for HD, the 95th percentile was calculated to reduce the influence of the outlier values. The formulas employed for the calculation of DSC, HD, Recall, and Precision are presented in Appendix A.

The utilized loss function consists of the weighted sum of the cross-entropy and DSC coefficient. Here, the loss function, 1 − DSC, was used. Cross-entropy quantifies the difference between two probability distributions for a given random variable. For the segmentation task, cross entropy is calculated in a pixel-wise manner and the average value is used as the result [34]. In this framework, cross-entropy enhances the training, especially in the first steps when DSC is not able to provide adequate information. In addition, because the two classes, referring to the left and right parotids, are almost equally represented with a similar number of voxels through the image, the simple form of cross-entropy without additional weights is utilized. The final loss is calculated through the sum of the two previously mentioned losses with equal weights of 1. The calculation formulas for the loss function are also presented in Appendix A.

### 2.4. Registration Expansion Framework

To further promote the clinical impact of this study, the DL segmentation method was expanded on a registration procedure based on the premise that significant anatomical alterations during RT necessitate treatment planning adaptation [5,6]. As such, we opted for automatic calculation of the parotid volume in the follow-up CBCT scans, utilizing the segmented masks from the pCT. To achieve this, a registration between the pCT as a moving image and the corresponding CBCT image as a target image was implemented.

#### 2.4.1. Registration Dataset

From the collected private dataset, additional low-dosage Cone Beam CT (CBCT) scans were acquired for 16 out of 23 patients over a span of 6 weeks. Each of the 16 patients underwent volumetric modulated arc therapy (5 sessions per week) with a 66 Gy dose delivered in 30 fractions.

#### 2.4.2. Registration Preprocessing

To achieve an accurate transformation of the initial segmentation masks to the CBCT of the target week, the following methodology was implemented. As such, due to the large difference in the acquisition parameters and intensities between the pCT and the CBCTs, we initially applied a registration step between the pCT and the CBCT1 (week 1) to move the labels in the CBCT1 space. Then, CBCT1 was used as the moving image to be transformed for the next week (week 5), allowing us to calculate the volumetric differences. In this regard, CBCT1 was registered to CBCT5 in the experiments where CBCT5 was used as the fixed (target) image. The only difference concerns the interpolation method for the segmentation masks, which was changed to nearest neighbors on account of existing labeling. As a preprocessing step, histogram matching was applied to all the pairs to be registered, thereby reducing the difference between the intensity distributions of the different image modalities. By minimizing the differences in the intensity distribution, the convergence of the registration was enhanced, especially in cases where images exhibit variations in illumination and contrast (CT-CBCT). Affine registration was utilized to align and match the pair of pCT-CBCT1 and CBCT1-CBCT5 by employing a combination of translation, rotation, scaling, and shearing transformations. This allowed us to further minimize the discrepancies between the corresponding points in the two images and reduce the burden for the subsequent deformable registration step.

#### 2.4.3. Registration Procedures

As mentioned above, registration included alignment and matching of pCT-CBCT1 and CBCT1-CBCT5 in distinct procedures. For the registration of pCT to CBCT1 images, the Advanced Normalization Tools (ANTs) registration library was utilized [35]. Registration parameters are presented in Table 3. Figure 2 presents the optimization procedure employed for the affine image registration. In detail, firstly, the affine transformation matrix is initialized. Then, the moving image is warped iteratively (using the current transformation) and the similarity metric (mutual information) between the fixed and warped moving images is calculated. An optimizer (regular step gradient descent) is used to update the transformation parameters to maximize this similarity metric (or minimize an error loss). This repetition continues until the convergence criteria are met, i.e., until the maximum number of iterations is reached (200 iterations) or the similarity metric converges. Finally, the transformation matrix and the warped moving image were generated, to be used as input for the deformable registration.

Regarding the different transformations, an initial rigid registration was utilized to capture potential rotations and translations in the pairs. Next, an affine registration step was implemented assessing global transformations (such as translation, rotation, scaling, and shearing). For both transformations, Mutual Information (MI) was used as the metric to be maximized during the optimization procedure, while a regularization value of 0.1 was selected to control the elastic deformation and preserve the smoothness of the structures. These procedures allowed an approximate alignment between the pairs. Following this, a deformable step based on normalized symmetric demons was employed, detecting more complex and local deformations. In the affine part of the registration, a random selection procedure for the tie points was employed, by including 10% of the total points of the fixed image. The same procedure was also used for the next step of the registration but at 25% of the total points. The denser sampling was used to improve the registration accuracy by including more details in the MI calculation (utilizing 32 bins).

For the registration of CBCT1 to CBCT5 images, a deformable registration methodology based on the fast symmetric demons registration from the SimpleITK library [36,37] was utilized for the initial affine alignment. The rationale behind this was to include correspondences as a hidden variable in the registration procedure. In this procedure, the Demons algorithm was used as global energy optimization, with the following regularization criterion employed as a prior to the smoothness of the transformation [37]. In this manner, flexibility between the overall representation is featured, allowing a small number of errors, rather than demanding optimal precision of point correspondences between image pixels (a vector field c). For the Symmetric Forces Demons algorithm, the standard deviation value was set to 3. The main equations for the Demons algorithm (for global energy and similarity) are presented in Appendix B.

In this work, the fast symmetric demons registration method [37] was implemented to evaluate the applicability of our segmentation method to the correct identification of volume changes in parotids and consequently to RT replanning necessity. To assess the registration framework efficiency, different metrics were calculated, including Mean Squared Error (MSE), Mutual Information (MI), and Normalized Cross Correlation (NCC) [38]. MSE indicates an error that should be reduced after registration, while MI and NCC constitute a measure of similarity that should be increased after a successful registration step. The calculation formulas are briefly presented in Appendix B. Furthermore, the volumetric percentage difference between the 1st and the 5th week of treatment sessions of the CBCT registration outputs was estimated and compared with the ground truth (i.e., the volumetric percentage difference of the CBCT1/CBCT5 images, annotated by experts). As such, the Mean Absolute Error (MAE) (used to measure the average absolute differences between predicted values and the actual values), the Root Mean Squared Error (RMSE) (used to measure the average magnitude of errors between predicted values and actual values), and the Pearson Correlation between the volumetric differences of the prediction and the ground truth were calculated to measure the strength of their relation.

### 2.5. Implementation Details

The proposed segmentation framework was implemented using PyTorch and the MONAI Python library [39]. The training and evaluation experiments were carried out on a Desktop with 64 GB RAM and Nvidia 4090 with 24 GB VRAM.

During the training, cropped boxes with dimensions (96, 96, 96) were extracted from the images by randomly selecting centers inside or outside the provided masks with probabilities of 80% and 20%, respectively. For the network’s training, the Adam optimizer with a learning rate of 0.0001 was applied. From each volume, four boxes were extracted each time. By utilizing smaller boxes throughout the training, GPU memory requirements were incredibly reduced, and the training was performed by effectively exploiting the available GPU VRAM. For the registration algorithms, the SimpleITK [36,37] and ANTs [35] libraries were utilized, whereas for the registration of CBCT1 to CBCT5 with SimpleITK, the standard deviation was set to 2 and the number of iterations to 600.

## 3. Results

### 3.1. Training and Validation Loss

To evaluate the performance of a model during the training process, the error or the difference between the model’s predictions and the actual target values in the training data (training loss), as well as the degree of the model’s effectiveness to generalize to new, unseen data (validation loss) was estimated. Figure 3 presents the learning curves of the proposed neural network model.

Training was performed for a constant number of epochs (600 epochs), which was selected by observing the trajectory of the loss function (reaching its final values approximately after 300 epochs). The final number of epochs was selected to achieve the same trajectory in all models and ablation experiments. To avoid overfitting, the model with the best DSC value in the validation set was saved for each experiment. Training and validation losses present a very close trajectory, which indicates that overfitting is mitigated, and the model can achieve high accuracy on unseen data.

### 3.2. Segmentation Comparison Experiments

In order to evaluate the model stability, the proposed segmentation DL network is compared with representative state-of-the-art segmentation models (including neural networks with residual connections, transformer-based neural networks, etc.), further emphasizing its performance. Due to the small size of the private dataset, the weights calculated for the public datasets were utilized. In this way, the pre-trained model from the public dataset was also tested for its ability to perform well in an external (private) set with different acquisition parameters. In Table 4, a short description of each implementation and the parameters used are provided to enhance the reproducibility of our results.

### 3.3. Segmentation Quantitative Comparison Analysis

The proposed methodology, along with the additional state-of-the-art segmentation networks presented above, are compared in terms of DSC (Overlap), Recall, Precision, and HD. Results of the 5-fold cross-validation for both the public and private datasets are presented in Table 5 and Table 6, corresponding to the public and the private datasets, respectively. Notably, the proposed AttentionUNet-based method achieved the best results in the two main metrics, while it performed comparably to the other two metrics in both datasets.

In the public datasets, UNETR achieved the lowest DSC, with a value of 78.82%, and the second highest HD, with 33 mm. Although SwinUNETR achieved a DSC of 81%, the HD value was pretty high (39.75 mm), making it unsuitable for the current application of radiotherapy. This could be explained by the fact that transformer-based networks require a large number of training samples to learn effectively to produce segmentations. UNet presented a low DSC value of 80.81, which indicates low ability to capture the details on the difficult-to-separate regions in the CT image, but achieved a small HD of 10 mm. SegResNet achieved the second-best results in most metrics while the proposed AttentionUNet-based method outperformed the others in the DSC value and HD value. In this regard, it produces segmentations with better overlap with the ground truth and also with the smallest distance (mean value of 6.24 mm and std of 2.47 mm).

Regarding the results on the private dataset (Table 6), the weights from the public dataset training were used as initialization for all the models, as well as the same cross-validation technique. Due to the differences in the acquisition parameters between the public and the private datasets and the small number of samples in the private dataset, evaluation metrics presented lower values in comparison. Recall values were very close for all models, while Precision was higher in SegResNet, indicating that less false positive voxels were estimated. Similar to the public dataset, the AttentionUNet-based method outperformed all other models in overlap and distance between prediction and ground truth, with higher mean DSC and the most stable metrics, as it can be extracted by the small values of the standard deviation. The HD metric was also small (5.35 mm) and stable. Although SegResNet performed comparably in the first three metrics, the large difference in the HD supports its lower prediction accuracy against the proposed AttentionUNet model.

### 3.4. Segmentation Qualitative Comparison Analysis

Segmentation examples of the proposed AttentionUNet network and the different comparison models are presented in Figure 4a for the public dataset and in Figure 4b for the private dataset. Regarding the public dataset, the AttentionUNet-based models achieved visually better results in comparison with the other DL models that produced slightly rotated masks in the same gland. As such, the proposed framework generated a smoother region and suppressed a fraction of the over-segmented regions of the other networks. By contrast, UNet, SwinUNETR, and SegResNet have over-segmented a small part of the parotid gland on the left and concurrently have produced fewer smooth masks. In a similar manner, SwinUNETR presented a larger difference in the parotid gland when compared with the ground truth.

Concerning the utilization of the private dataset as an external validation, it can be observed that the proposed AttentionUNet model best approximates the ground truth as it reduces over-segmentation, while also producing smoother segmentations. In comparison, UNet failed to outline a large portion of the parotid gland, whereas the other models produced slightly different structures on the left and under-segmented the right parotid. In this example, although AttentionUNet did not capture the whole structure on the right, it reduced false positives and produced very accurate results on the left one.

### 3.5. Results on the Registration Expansion Framework

Following the segmentation framework, the registration feasibility study was implemented. In Figure 5, an example output from the registration process is presented. Moving refers to the image to be aligned with the fixed image. Transformed image refers to the results of the registration process. In the final two images of each row, the edges of the fixed image (reference) are presented on top of the moving and moved images in order to qualitatively assess the performance of the utilized registration method according to the edges of the image. The first row shows an example of the first registration step, where the pCT image is registered to the CBCT1 (week 1), while the second row shows an example of the second registration step, where the CBCT1 is registered to the CBCT5 (week 5). In Figure 5a, the pCT and CBCT images present a larger difference and the number of slices is not the same. Due to this fact, moving and fixed images are far apart and the edges of the fixed image do not match the edges of the moving image. However, registration manages to bring them to approximately the same space, which can be seen both in the similarity between the fixed and the transformed image and in the alignment of the edges of the fixed image when they are superimposed on the transformed image. In Figure 5b, an example pair of the registration between CBCT1 and CBCT5 shows the smaller difference between them before registration, as well as the ability of the second registration step to align images and their edges.

As presented in Figure 5a,b, the transformed image is more similar to the fixed one, whereas the edges of the fixed image are closer to the real edges in the transformed image. On the moving image, it can be observed that the edges of the fixed image initially do not match the real edges of the moving one, while after registration, the edges of the fixed image are aligned with the transformed image.

In Table 7, the values of the metrics for the assessment of the registration between the two images are presented. Mean values of the Mean Squared Error (MSE), Mutual Information (MI), and Normalized Cross-correlation (NCC) are calculated for the pairs of images before (fixed and moving images) and after (fixed and transformed) registration and are presented for both tasks (i.e., pCT to CBCT1 and CBCT1 to CBCT5).

The calculated metrics support the registration accuracy, as the MSE is reduced and CC and MI are increased in both tasks. Of note is that the initial step of pCT to CBCT1 was performed in order to enhance the step of CBCT1 to CBCT5, which was the target of this investigation, providing indices for required RT replanning. As expected, the registration between pCT and CBCT1 was more challenging, confirmed by the large MSE before the registration. After the registration, MSE was reduced and the similarity measures were increased, indicating effective alignment between the different modality images. In the second task, MSE was significantly reduced from 94 to 16, MI was increased from 46 to 66, and NCC from 89 to 96, further demonstrating successful registration.

Subsequently, we calculated the volumetric percentage difference between the first week and fifth week of RT, based on the premise that significant volumetric alterations during RT or between RT sessions necessitate planning adaptations. As such, the parotid volume difference between CBCT1 and CBCT5 as a result of the registration tasks was estimated and compared to the ground truth (i.e., the volume difference as annotated by experts). Table 8 presents the MAE, the RMSE, and the Pearson Correlation Coefficient (PCC) of the volumetric percentage difference between the registration output and the ground truth of the CBCT1/CBCT5 images.

The volumetric percentage difference demonstrated very small values for both MAE and RMSE, accounting for limited divergence from the ground truth. In fact, the 7.577 value of MAE suggests that the mean percentage difference deviation between the experts’ annotation and the registration output was less than 8%. Moreover, the results show a statistically significant positive correlation coefficient of 0.723 (*p*-value < 0.002), which indicates that the variables move in the same direction with relatively high correlation.

## 4. Discussion

Automated segmentation in HN cancer is of great significance for RT planning, as it can reduce time and provide crucial information on the anatomical structures of the region of interest. In this study, a DL segmentation framework is proposed, aiming for fast, automatic, and accurate delineation of the parotid glands. Although automated systems for enhancing adaptive radiotherapy exist (e.g., Varian ETHOS, RT-Plan, etc.), we propose a low-requirement AI-based decision support system that can provide insight and effectively assist radiotherapy planning [45,46]. The fact that our study is highly connected to the current clinical workflow combined with further extensive evaluation experiments and technical improvements could provide a tool applicable to real clinical procedure. In this regard, the utilized AttentionUNet architecture achieved high overall performance in both public and private datasets, supporting the potential applicability of the proposed method to the clinical workflow.

Regarding the proposed DL model, the AttentionUNet architecture utilized in this study exploits the addition of an attention mechanism to the UNet structure [31]. As such, the UNet-like structure with five layers in the encoder enables the network to extract useful context information in the encoder path and then reconstruct the input and the spatial information to produce the final mask in the decoder. In addition, by including learnable weights, the network learns to focus on the regions inside the image that are more relevant to the task, in this case, the parotid glands. In this way, false positive regions are effectively suppressed, especially in the CT images where the structures are not easy to separate [47]. Moreover, the integration of attention gates in the model’s design allowed us to incorporate features from deeper encoder layers that introduce better context information in each layer of the decoder [48]. Further enhancement of the segmentation accuracy (especially in the early epochs, the weights could not be pushed in the right direction) was accomplished with a combination of dice and cross-entropy loss.

For a robust and unbiased evaluation of the segmentation framework, the segmentation accuracy was tested on a unified public dataset (consisting of two public subsets) and a private dataset. By including images with different acquisition parameters and settings, we aimed to validate the generalizability of our method, enhancing the model’s ability to work effectively under different clinical protocols [49]. As such, the model demonstrated high performance on the public dataset by achieving DSC and HD values of 82.65 and 6.24, respectively. The private dataset was also used as an external validation dataset, where delineation was much more difficult (with variations in the acquisition parameters in comparison with the public training datasets further complicating training tasks). However, the proposed method presented satisfying results that support its ability to generalize to other datasets. As such, the proposed model achieved DSC of 78.93 and HD of 5.35 in the private dataset. In this setting, the pre-trained model resulting from the training in the public dataset was applied in a 5-fold cross-validation in the private set, also achieving high metrics, albeit with a small reduction deemed inevitable due to the large difference in the acquisition parameters among the datasets [50]. It should be noted that the proposed AttentionUNet-based network outperformed many state-of-the-art DL segmentation methods, both in terms of overlap between the predicted and the ground truth masks and in terms of the distance of their edges. On this premise, HD is required to be as low as possible due to its applicability in RT (where the distance between the true region and the delineated regions should be short, to increase tumor coverage and alleviate OAR overdose) [51].

Subsequent to the segmentation framework, the predicted masks from the pCT of the private dataset were further utilized in a registration schema, assessing the method’s potential use in clinical practice for RT planning adaptations. To achieve this, two image registration tasks were designed and implemented in order to transfer the predicted masks to the CBCT of week 5 of the radiotherapy workflow. The ANT registration method was utilized to align the pCT with the CBCT1, and the calculated transformation was applied to the labels of the parotids to align the labels to the CBCT1. This was an intermediate step, because the distance between pCT and CBCT of week 1 was smaller than that of week 5. This step allowed the registration to focus on the large differences in the acquisition parameters without introducing large differences in the patient’s anatomy during the RT treatment. In the second task, CBCT1 was aligned to CBCT5 (through Fast Symmetric Demons registration) to transform the labels of the parotid glands and calculate their volume in week 5. Of note is that CBCT images were more similar to each other in terms of the acquisition parameters and the intensity values, which in turn enhanced the registration procedure. As presented in the registration results section, the proposed workflow was able to approximate the volumetric changes of the parotid glands with high accuracy. Interestingly, the volumetric percentage difference between CBCT1 and CBCT5 indicated a 7.55 MAE when compared to the experts’ observations. This indicates the high performance of the registration process, taking into account that variations in the registration of cancer CT scans can also occur among different observers. In fact, observers may have individual techniques, preferences, or interpretations when performing registrations, leading to differences in how they align or match the images [21]. The use of automated registration tools can contribute to reducing variability among observers.

### Limitations and Future Recommendations

Some considerations should be given when interpreting the results of this study. First and foremost, this study included a relatively small sample data size. Although three datasets were combined, the significant variance among patients could limit the generalizability of the findings. As such, the framework proposed in this study seems to lack the requisite robustness for practical deployment in real-world scenarios. The performance metrics and results (although encouraging) suggest limitations in the model’s ability to generalize to diverse and complex scenarios for real-world variations. In addition, the segmentation and subsequent registration processes were applied to delineate only the parotid gland, therefore training for shape constraints of small organs and leaving out larger OAR shape characteristics. This might introduce biases and limitations when expanding to other organs. Moreover, the study focused on segmentation and registration algorithmic performance, omitting potential limitations associated with their implementation in clinical practice, such as computational resources, training time, and integration with existing medical imaging systems. As a result, caution should be exercised before deploying the DL model in real-world settings, and further refinements or improvements may be necessary to support the adoption of automated DL segmentation systems. In this regard, we aim to expand our framework to involve additional patient cohorts and expert observations to enhance its efficacy and reliability.

## 5. Conclusions

In this study, a novel DL neural network framework (AttentionUNet) is implemented for the automated segmentation of parotid glands in HN cancer. The model is evaluated across two publicly available datasets and one private dataset, and compared against other state-of-the-art DL segmentation approaches. The proposed model demonstrates high segmentation results with a Dice Similarity Coefficient of 82.65% ± 1.03 and a Hausdorff Distance of 6.24 ± 2.47, outperforming the other DL methods. Following this, the segmentation process is expanded, applying to the segmentation output to align pCT and CBCT images. The registration results exhibit high similarity when compared to the ground truth, offering valuable insights into the impact of radiotherapy treatment for adaptive treatment planning adjustments. The implementation of these proposed methods underscores the efficacy of DL and its potential applicability in the clinical workflow of image-guided RT for applications such as the identification of the volumetric changes in OARs.

## Figures and Tables

**Figure 1 bioengineering-11-00214-f001:**
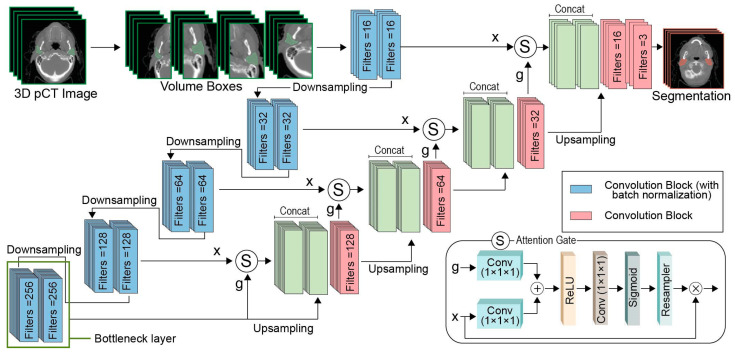
The architecture of the AttentionUNet-based method. The attention gate block is presented in the bottom right.

**Figure 2 bioengineering-11-00214-f002:**
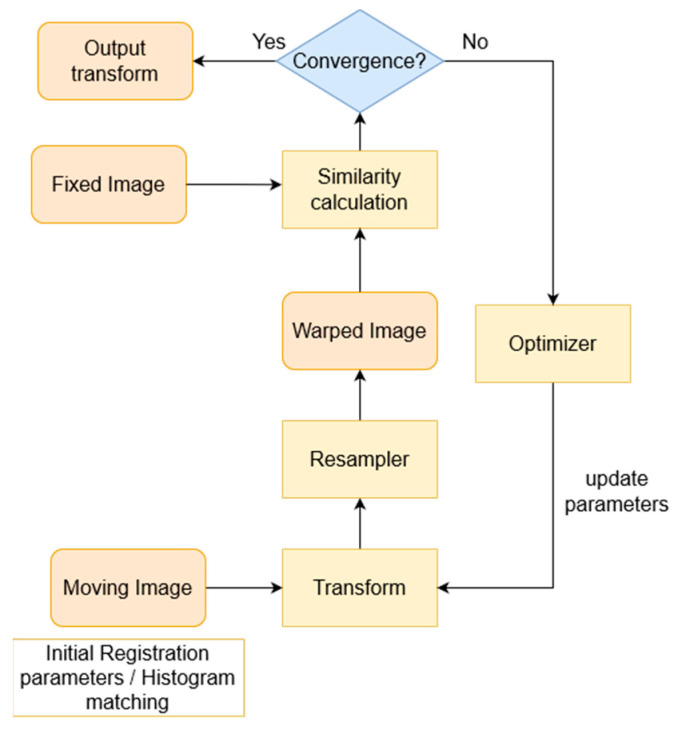
Affine registration flowchart implemented for the pCT to CBCT1 and CBCT1 to CBCT5 steps.

**Figure 3 bioengineering-11-00214-f003:**
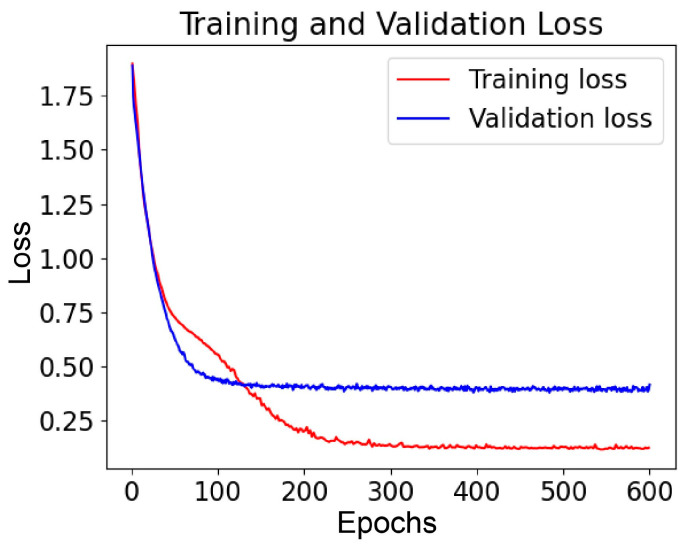
Training and validation loss for 600 epochs.

**Figure 4 bioengineering-11-00214-f004:**
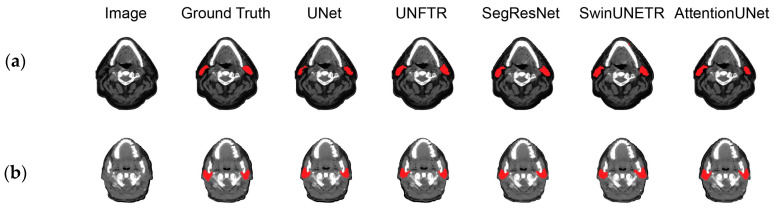
Segmentation results of the tested models on (**a**) the public dataset; (**b**) the private dataset. Each column represents different algorithmic designs. The segmentation masks of the parotid glands are displayed in red.

**Figure 5 bioengineering-11-00214-f005:**
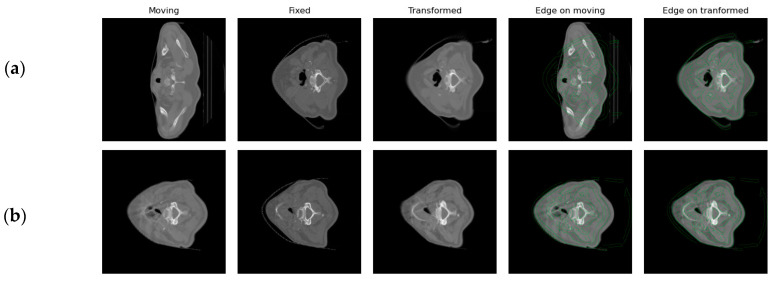
An example of the registration processes for (**a**) pCT to CBCT1; (**b**) CBCT1 to CBCT5.

**Table 1 bioengineering-11-00214-t001:** Architecture sizes of the encoder and the bottleneck layers.

Layer Blocks	Input Size	Output Size
Encoder Block 1	[1, 96, 96, 96]	[16, 96, 96, 96]
Encoder Block 2	[16, 96, 96, 96]	[32, 48, 48, 48]
Encoder Block 3	[32, 48, 48, 48]	[64, 24, 24, 24]
Encoder Block 4	[64, 24, 24, 24]	[128, 12, 12, 12]
Encoder Block 5	[128, 12, 12, 12]	[256, 6, 6, 6]

**Table 2 bioengineering-11-00214-t002:** Architecture sizes of the decoder layers.

Layer Blocks	Input Size	Output Size
Decoder Block 1	[256, 6, 6, 6]	[128, 12, 12, 12]
Decoder Block 2	[128, 12, 12, 12]	[64, 24, 24, 24]
Decoder Block 3	[64, 24, 24, 24]	[32, 48, 48, 48]
Decoder Block 4	[32, 48, 48, 48]	[16, 96, 96, 96]
Decoder Block 5	[16, 96, 96, 96]	[3, 96, 96, 96]

**Table 3 bioengineering-11-00214-t003:** ANT registration parameters.

Parameter	Value
Interpolation	linear
Intensity clipping	[0.005, 0.995]
Histogram-matching	Yes
Initial transform	Rigid [0.1]
Initial transform metric	MI (32 bins)
Iterations	50 × 50 × 25 × 10
Minimum convergence	1.00 × 10^−6^
Second transform	Affine [0.1]
Second transform metric	MI (32 bins)
Iterations	100 × 70 × 50 × 20
Transform	SyN
Transform metric	Cross-Correlation
Iterations	100 × 70 × 50 × 20

**Table 4 bioengineering-11-00214-t004:** Segmentation comparison models.

Models	Description	Parameters
UNet [40,41]	Fully convolutional neural network. Consists of an encoder path, which extracts the image’s context, and a decoder path, which reconstructs the spatial information. Skip connections from the different levels of the encoder to the corresponding levels of the decoder transfer useful information between them.	Encoder layers: 16, 32, 64, 128, 256; decoder layers are inversed
SwinUNETR [42]	Exploits the transformer blocks to model long-range dependencies in an image. It includes a Swin transformer encoder to capture features in different resolutions, using shifted windows in the self-attention module. The decoder part is connected via skip connections to the encoder and consists of convolutional blocks.	Features dimension: 48; number of heads: 3, 6, 12, 24
UNETR [43]	Architecture refers to a UNet-like structure with encoder/decoder parts and additional transformer blocks in the encoder blocks replacing convolutions.	Hidden layer dimension: 768; attention heads: 16; feedforward layer dimension: 2048
SegResNet [44]	Includes a larger encoder to extract useful information and a smaller decoder to produce the segmentation mask. It is based on the MONAI library without the use of the Variational Autoencoder. The main construction block of the encoder is ResNet blocks containing residual connections.	Number of encoder layer filters: 8, 16, 32, 64, 128; kernel size of three; group normalization

**Table 5 bioengineering-11-00214-t005:** Results on the public concatenated dataset. Mean values of DSC, Recall, Precision, and HD are presented.

Method	DSC (%)	Recall (%)	Precision (%)	HD (mm)
UNet	80.81 (±0.66)	**84.10** (±1.02)	80.28 (±1.21)	10.76 (±2.10)
SwinUNETR	81.02 (±1.93)	81.55 (±0.79)	83.40 (±2.61)	39.75 (±15.37)
UNETR	78.82 (±0.91)	82.12 (±0.21)	78.53 (±1.26)	33.06 (±5.44)
SegResNet	81.75 (±1.00)	81.11 (±2.16)	84.73 (±0.98)	12.07 (±4.93)
AttentionUNet	**82.65** (±1.03)	82.66 (±2.10)	**84.77** (±0.76)	**6.24** (±2.47)

Standard deviations from the cross-validation are placed inside the parentheses. **Bold** indicates the overall best result for each metric.

**Table 6 bioengineering-11-00214-t006:** Results on the private dataset. Mean values of DSC, Recall, Precision, and HD are presented.

Method	DSC (%)	Recall (%)	Precision (%)	HD (mm)
UNet	77.11 (±3.57)	80.07 (±4.13)	76.59 (±2.46)	5.78 (±0.44)
SwinUNETR	74.46 (±5.53)	**80.44** (±3.70)	73.45 (±6.75)	30.46 (±19.37)
UNETR	74.12 (±6.62)	75.08 (±6.13)	75.38 (±5.45)	52.91 (±26.82)
SegResNet	78.54 (±4.30)	77.55 (±7.20)	**81.63** (±1.07)	16.92 (±19.98)
AttentionUNet	**78.93** (±3.37)	79.99 (±3.78)	80.04 (±1.82)	**5.35** (±0.42)

Standard deviations from the cross-validation are placed inside the parentheses. **Bold** indicates the overall best result for each metric.

**Table 7 bioengineering-11-00214-t007:** Results on the registration tasks.

Task	MSE	MI	NCC
pCT-CBCT1 (before registration)	468.52353	0.1752	0.5785
pCT-CBCT1 (after registration)	306.0259	0.4638	0.8116
CBCT1-CBCT5 (before registration)	94.1346	0.4684	0.8939
CBCT1-CBCT5 (after registration)	16.7582	0.6675	0.9681

**Table 8 bioengineering-11-00214-t008:** Volumetric percentage difference.

Method	MAE	RMSE	PCC
CBCT1→CBCT5	7.577	9.256	0.723

## Data Availability

The data presented in this study are available on request from the corresponding author. The data are not publicly available due to privacy reasons.

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
