# Peer review of "Towards Automation in Radiotherapy Planning: A Deep Learning Approach for the Delineation of Parotid Glands in Head and Neck Cancer"

_bioengineering, 2024, doi:10.3390/bioengineering11030214_

Round 1
Reviewer 1 Report
Comments and Suggestions for Authors
This is a well written and well performed report on the utility of a deep learning approach in the planing of radiotherapy in parotis gland cancer. The topic is interesting and important. The authors describe the procedure in detail and check the performance of their new model by established parameters. Their conclusion, that the implementation of these methods underscores the efficacy of deep learning and its potential applicability into the clinical workflow is justified.
The only thing I would like to add is the point, that, given the data in this paper, their new neural network framework is by now not good enough to be applied in a real life scenario.
Reviewer 2 Report
Comments and Suggestions for Authors
This paper presents an deep-learning-based parotid gland segmentation method for head and neck cancer. The method is sound. However, there are some concerns about this study.
- It is unclear what is the main challenge in using deep learning to delineate the parotid glands in head and neck cancer.
- It is better to well summarize the contributions in the introduction.
- Please briefly indicate the main difference of the proposed method to those in the related studies.
- More studies in medical image analysis based on deep learning needs to be cited.
- In Figure 3, it is not well easy to understand how to observe this figure.
- In the model training, is the early stopping used ?
- There are writing problems, e.g. grammatical errors.
Minor refinement
Reviewer 3 Report
Comments and Suggestions for Authors
Although the article is very well written, the following points need to be addressed in the revision:
- The abstract is adequate in length and structure. However, the following minor changes are suggested to further improve it:
- Line No. 19: correct the use of “model’s”
- Line No. 22: rephrase/ grammatical correction “aligning planning CT and CBCT images”
- Line No. 23: correct the use of “other DL method”
- Line No. 23-24: 82.65 and 6.24 must accompany standard deviation with a ± symbol (5 fold mentioned in the work)
- Line No. 25: rephrase to improve the script and avoid repetition in “treatment for treatment planning adaptations”.
- It is unusual to mention results in the introduction as given in Lines 93-95.
- Add the contributions of your work in bulleted form at the end of the introduction followed by the organization of the article.
- Page 3, Line 138: “training procedure” should be replaced by “training progress”.
- Please thoroughly check your article for typos and grammatical mistakes.
- In preprocessing, the “spacing of (0.87, 0.87, 3)” is not clear concerning Line 126 (page number 3). Add some text to the article to make it more elaborative.
- Why you are providing a range for several slices (Line numbers 125, and 132, on page number 3)? How you tackle it with 5-fold cross-validation.
- In the architecture of a-unet, I could not find the bottleneck layer in Figure 1 as mentioned in Line Number 174. Highlight it in Figure 1.
- In the architecture, referring to Line numbers 184-185, recheck the bottleneck concept for auto-encoders and u-nets.
- I could not find the outcomes/sizes during upsampling or downsampling. This is essential to mention in the form of a table or in the text to see the proper configuration.
- Line 208-209: the masks are to be fed by default. You can also mention the type of mask by correcting the sentence.
- Section2.4: The registration procedure through affine transformation is not clear. I suggest adding a flowchart and showing what procedure you adopted.
- Line 322: I think the stability is achieved after 300 epochs. Please double-check it in Figure 2.
- In Figure 2, I feel the training and validation curves have been altered in labels. Check the professional trends.
- How did you find the tie points for the registration?
- In Table 5, I could not find “Correlation” defined in the text part of the article.
- Add a subsection, Limitations and Future Recommendations, before the conclusions section to help readers find your work important.
Moderate level English language revision is required.
Round 2
Reviewer 2 Report
Comments and Suggestions for Authors
No further question.
Comments on the Quality of English LanguageMinor revision
Author Response
Thank you. We have edited English language to the best of our abilities.
Reviewer 3 Report
Comments and Suggestions for Authors
All the points have been thoroughly addressed. The following minor change is still recommended (as point 3 in the earlier list of reviewer comments):
1. The article's organization is missing at the end of the introduction section.
Author Response
We would like to thank the reviewer for their detailed and thoughtful comments on our submission. We have revised the manuscript accordingly.
- The article's organization is missing at the end of the introduction section.
Answer: Thank you for pointing this. We missed the comment in our previous revision. He have added the organization at the end of the introduction section.
Added text. “The subsequent sections of the paper are organized as follows: Chapter 2 provides a comprehensive overview of the datasets and the DL methodological framework for segmentation and registration. In Chapter 3, the results of the proposed schema are separately presented for segmentation and registration are presented . The implications of our methods and results are discussed in Chapter 4. The entire paper is summarized in Chapter 5.”